# Gulls as Indicators of Environmental Changes in the North Atlantic: A Long-Term Study on Berlenga Island, Western Portugal

**Filipe R. Ceia** [1,*], **Nathalie C. Silva** [2], **Vitor H. Paiva** [1], **Lurdes Morais** [3], **Ester A. Serrão** [2] **and Jaime A. Ramos** [1]

1 University of Coimbra, MARE—Marine and Environmental Sciences Centre / ARNET—Aquatic Research Network, Department of Life Sciences, 3000-456 Coimbra, Portugal; jramos@uc.pt (J.A.R.)
2 University of Algarve, Algarve Centre of Marine Sciences (CCMAR), Campus Gambelas, 8005-139 Faro, Portugal; eserrao@ualg.pt (E.A.S.)
3 Institute of Nature Conservation and Forests (ICNF), 1050-191 Lisboa, Portugal
* Correspondence: ceiafilipe@zoo.uc.pt

**Abstract:** In recent decades, the breeding populations of the yellow-legged gull *Larus michahellis* (YLG) have increased significantly, primarily due to the increase in open refuse dumps and discards from fisheries. Portugal's largest YLG breeding colony is located on Berlenga Island, where population numbers have been monitored since 1974. The population grew exponentially until 1994, prompting the implementation of population control measures, including culling adult birds and eggs. A long-term data base including number of breeding birds (since 1974), breeding parameters (since 2002), and trophic niches (using stable isotopes since 2011) of YLG breeding on Berlenga was related with oceanographic parameters, fish landings and quantity of urban waste. Trophic ecology showed strong relationships with oceanographic parameters (wNAO, Chl-a, and SST) and fisheries landings (the 10 most frequently consumed species by the YLG, traded at fish auctions in the main fishing harbour nearby). The results indicated significant relationships between reproductive performance and fisheries landings, particularly with demersal species that gulls primarily access through fisheries discards. However, population control measures played a pivotal role in stabilising and even reducing the population, despite sporadic events of poor oceanographic productivity in the past decade having a consistent impact on the reduction in breeding individuals.

**Keywords:** yellow-legged gull; isotopic niches; feeding ecology; *Larus michahellis*; stable isotopes; breeding performance; census; oceanographic conditions; NAO

## 1. Introduction

Gulls are social seabirds that tend to form extensive colonies during their breeding seasons. Due to their role as opportunistic scavengers and apex predators, often reliant on both human-related activities and marine natural prey, gulls can be used as bioindicators of anthropogenic effects and shifts in the marine environment [1–4]. By delving into the dietary habits of gulls, the dynamics of marine resource availability can be monitored, which in turn may reflect long-term variations in the environment [5]. Therefore, scrutinising the dietary habits of gulls facilitates the assessment of local prey distribution and long-term marine ecosystem shifts, and elucidates the birds' responses to oceanographic changes [6,7]. Moreover, it is essential to understand the impact of predictable anthropogenic food subsidies, such as fisheries bycatch and food available in open-air landfills, on wildlife and assess their extent. This understanding is pivotal for sustainable resource management [8].

In Europe, some populations of gulls have shown a high increase over the last decades, prompting concern due to the impacts on ecosystems and urban areas, such as changes in vegetation cover and plant composition, competition with other species, or interaction with humans [9–12]. The increase in large gull populations (*Larus* spp.) has been documented since the early 20th century [13,14]. This has been closely linked to the proliferation of open refuse dumps and the rise in discarded bycatch from industrial fisheries [7,15–18].

Multiple investigations have been conducted to explore the spatio-temporal resource utilisation patterns of gulls in order to enhance our comprehension of their feeding ecology, e.g., [19–22], including the discernment of alterations in ecosystem dynamics through food web structural modifications [23–25]. These studies have revealed significant insights into the foraging behaviour, dietary preferences, and ecological roles of gulls within their respective habitats, such as adaptations to changes in prey availability. Additionally, these investigations have provided valuable insights into the potential role of gulls as bioindicators to assess environmental shifts and anthropogenic influences on coastal ecosystems. Nonetheless, there remains a lack of comprehensive, long-term studies that simultaneously correlate their population abundance, reproductive success, oceanographic conditions, availability of anthropogenic resources (arising from fisheries and refuse) and feeding ecology.

The yellow-legged gull *Larus michahellis* (YLG) is a large and generalist gull species. Their breeding distribution spans from northern Africa and southern Europe to Macaronesian regions [26]. In general, YLG populations have grown significantly over the last decades [10,16], attributed to heightened resource availability originating from human activities [7,27]. Like other large gulls, YLGs frequently supplement their diet with anthropogenic food sources, such as those found in refuse dumps, landfills and discards from fisheries [11,28–31]. This can have an impact on local biodiversity, affecting individuals, populations, and the overall structure and functioning of entire ecosystems [8]. The exploitation of predictable anthropogenic food subsidies by opportunistic species enhances their breeding success by increasing breeding investment, chick growth rates, hatching success [15], body condition [32], and survival [33].

In Portugal, the largest YLG breeding colony is located on Berlenga Island, where population data have been systematically monitored since 1974. Historical records indicate approximately 1000 breeding pairs in 1939 [14], followed by around 1300 breeding pairs in 1974 [34]. Afterwards, a notable exponential population increase was observed between 1974 and 1994, growing at a rate of 16% annually, culminating in roughly 22,000 breeding pairs [35]. From 1994 to 1996, the Institute for Nature Conservation and Forests (ICNF) allowed for the culling of adult birds, significantly reducing the breeding population by half in 1996 [35]. Since then, a population control strategy was implemented, encompassing the culling of eggs throughout the island during the breeding season, with the exception of two designated control areas, representing less than 1% of the population [36]. Population numbers remained relatively stable until 2009, after which a gradual decline became apparent. As of 2022, the YLG population had dwindled to around 2200 breeding pairs [36].

Yellow-legged gulls from Berlenga Island feed largely on fishery discards sourced from Peniche harbour (13 km away) [21]. Several commercially mid-trophic level fish species, traded within Peniche harbour, are of particular importance in their diet [21,37,38]. As a generalist species, YLG exhibits a wide spectrum of prey preferences and habitat use. However, during the breeding season, they are central-place foragers, limiting their access to resources in the vicinity of their colony [21]. Thus, their marine food sources are subject to availability fluctuations, with an observable increase in consumption of refuse tips when the accessibility of fish and other marine prey is restricted [39,40]. This species can travel up to 100 km in search of food [21]. Yet, longer trips might negatively influence their breeding performance, considering the impact of increased foraging time on the time spent at the nest [41]. Furthermore, the reliance on anthropogenic resources is directly related to its distance from the breeding site; i.e., the closer the resource, the greater its exploitation [7,27,37]. Nonetheless, even when the colony is situated in close proximity to anthropogenic sources like landfills, these gulls tend to favour a marine foraging strategy during the breeding season [40] and to heavily rely on marine prey [38]. Exhibiting remarkable flexibility both at the individual and population levels, YLGs frequently vary between marine, mixed and terrestrial foraging strategies [40]. This adaptability is especially evident when marine resources are scarce, and oceanographic conditions prove unfavourable,

leading to a heightened frequency of terrestrial foraging behaviours and a diet based on anthropogenic-derived resources [40].

The North Atlantic Oscillation index (NAO) represents a large-scale atmospheric pattern of variability that affects the mid and high latitudes of the Northern Hemisphere, thereby triggering cascading effects throughout the North Atlantic region [42]. This, in turn, has considerable impact on marine ecological processes such as phytoplankton productivity and fish stocks [43,44]. Such shifts may thus be reflected in the realm of marine resource availability, directly impacting seabird populations, including gulls, that are affected by a bottom-up effect, instigated by changes in prey abundance and distribution within the North Atlantic [40,45]. Recent empirical data provide evidence of the ecological impacts of extreme climate events in shaping key life-history traits of seabirds when facing adverse environmental conditions. An example of this phenomenon is evidenced by the contrasting environmental conditions observed during extreme negative (2010) and extreme positive (2015) phases of the NAO index—representing the most negative and positive phases, respectively, within recent decades [46].

Under limited resources, the condition of adult birds will be poorer, reducing, for instance, the number of breeding individuals, egg clutches, egg volume, hatching success, fledging success, and overall reproductive performance [17,25]. The reproductive success is intricately related to the diet; thus, the quality of nourishment acquired during pre-laying and incubation stages can be reflected in egg clutches, egg volume, and hatching success [47–49]. The quality of eggs of herring gulls *Larus argentatus* is affected by dietary and environmental factors [50]. In particular, diminished egg volume in *L. argentatus* correlated with decreased productivity in the Great Lakes region of North America, concomitant with a decline in the quality and abundance of prey [23,51]. Moreover, a long-term study elucidated substantial shifts in the egg clutches and egg volume of black-legged kittiwakes *Rissa tridactyla* in response to oceanographic conditions, specifically variations in the influx of warm Atlantic Water into the Barents Sea [52]. This relationship between food quality and breeding performance is also observed in YLG, as shown experimentally for Illas Cíes in NW Spain, by applying the food supplementation of parents during the egg formation phase [53]. This study demonstrated the influence of parental nutritional condition on egg volume and hatching success, revealing greater effects in nests where supplementary food was provided to adults.

The aim of this study was to explore the relationship between environmental factors, trophic ecology, the number of breeding birds, and the reproductive success of a large colony of yellow-legged gulls breeding on Berlenga island, Portugal. To this end, a long-term dataset including population numbers (through annual census, since 1974), breeding parameters (including hatching success and egg parameters, since 2002), and trophic niches elucidated through stable isotopes ($\delta^{15}$N and $\delta^{13}$C, since 2011), was correlated with oceanographic parameters (Chlorophyll-a concentration, sea surface temperature and the extended winter North Atlantic Oscillation Index), fishery landings (species landed in the main fishing harbour in the area, Peniche harbour) and urban waste records (from residues delivered to the proximate active landfill and non-differentiated residues collected in the city of Peniche and for the broader West region). This study aims to evaluate the ecological dynamics of YLGs concerning both natural and anthropogenic environmental factors, including population control measures, and specifically to address the following objectives:

1.　Investigate the relationships between long-term fluctuations in the annual breeding population size of YLGs and the large-scale variations in oceanographic conditions, fisheries landings, and urban waste.
2.　Examine the associations between the long-term annual reproductive performance and trophic niche of YLGs with the large-scale variations in oceanographic conditions, fisheries landings, and urban waste.

We hypothesise that YLG population dynamics on Berlenga Island are influenced by variations in environmental factors, despite implemented population control measures. We

also predict that the reproductive performance and trophic niche of YLGs will be influenced by the quality of oceanographic conditions, with better conditions associated with higher hatching success and narrower trophic niches. This presumes that changes in environmental factors, such as oceanographic conditions, fisheries landings, and urban waste can affect the annual reproductive performance and trophic ecology of the gull population.

## 2. Materials and Methods

### 2.1. Study Area

The Berlengas Archipelago is a Marine Protected Area (MPA) and consists of three groups of islands: Berlenga Grande (i.e., Berlenga Island), Estelas, and Farilhões. Moreover, bird data were collected on Berlenga Island, Portugal (39°24′ N, 009°30′ W) (Figure 1). Berlenga is a small rocky neritic island of ca. 78.8 ha, located within the continental shelf about 11 km off western Portugal coast, surrounded by shallow waters. The Berlengas Archipelago features high marine productivity due to upwelling, more intense from April to September, fostering heightened concentrations of chlorophyll-*a* and low sea surface temperatures [54].

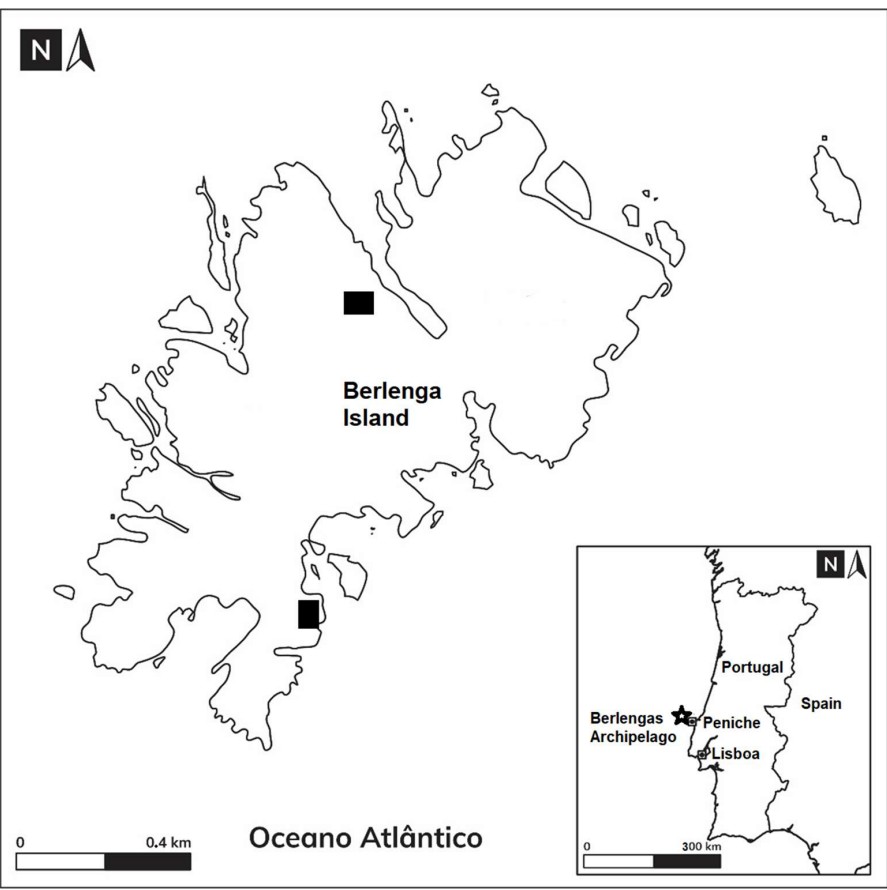

**Figure 1.** Berlenga Island, west central Portugal indicating the city of Peniche and the Berlengas Archipelago. The star indicates the study site, and black rectangles indicate control areas (see Section 2).

### 2.2. Sampling Procedures

Individuals of YLG within the colony were bred, their egg clutches (i.e., number of breeding individuals and total number of eggs) were counted during the annual census, and population control campaigns for YLGs were carried out by the Institute for Nature Conservation and Forests (ICNF), which started in 1974 (census of breeding birds) and in 1999 (counts of the eggs). The census of breeding birds is conducted by two ICNF wardens and starts approximately 1.5 h before sunset, during two consecutive days, at which point the majority of birds have already returned to the colony. On one day, the birds present

on the slopes of Berlenga are counted, and on the other day, those on the plateau [55]. The census takes place at the beginning of the incubation period (i.e., early May).

Egg culling has been carried out on Berlenga Island since 1999, and campaigns are scheduled based on the timing of reproduction (egg laying—approximately late April to mid-June; incubation—around 28 days), the size of the colony, the maximisation of efficiency, and minimising costs. The participants (ICNF wardens) have prior experience to avoid missing nests. Seven consecutive days are required for the entire island to be thoroughly surveyed with a team of 8 wardens. The entire island is traversed on foot, with the exception of inaccessible cliffs and two designated permanent control areas, each spanning 800 m$^2$. The two control areas are visited every 3 days; one was located on the slope, and the other on the plateau (Figure 1), for the assessment of annual breeding success [36]. Given the nature of YLGs replacing their clutches and the asynchronous nature of the initial egg laying, ICNF wardens undertake three separate egg-culling campaigns. The three campaigns are spaced at 21-day intervals (between May and June), to ensure that no new gull chicks hatch in the meantime and to allow gulls, who had their eggs culled, to replace them (approximately 30% of gull pairs can replace their eggs once or twice).

The information collected during the annual census and population control campaigns includes the number of breeding individuals (BI), the total number of eggs culled (total eggs), the number of eggs per breeding pair (estimated, i.e., total eggs pair$^{-1}$ = total eggs/(BI/2)), the percentage of eggs hatched in control areas (hatching success) and total days from the year's commencement to the date of the first egg laid per breeding season within control areas (first egg). Additionally, to ascertain mean egg volume of the clutch (egg volume), only nests with three eggs (the dominant class size) were randomly selected (sample size varied from 20 to 30 nests) in the control areas since 2012 (except in 2013 and 2020). We measured egg length (L) and egg width (W) using digital callipers (to the nearest 0.1 mm) and calculated egg volume (in cm$^3$, a mean value per nest to counter pseudo-replication) using the formula: L × W$^2$ × 0.476 [56]. The breeding parameters (total eggs pair$^{-1}$, hatching success, first egg and egg volume) were used as measures of the gulls' breeding performance during different years in relation to their trophic ecology, as well as oceanographic conditions, fisheries landings, and urban waste.

During 2011–2022 (except in 2019 and 2020), a total of 168 breeding individuals were sampled for approximately two weeks in May/June, when adults were incubating. By using square traps over the nests, adults with three-egg clutches were captured and ringed. The nests were chosen randomly across the entire island (excluding control areas), and individuals were not resampled across years.

### 2.3. Stable Isotope Analysis (SIA)

Stable isotope analysis (SIA) has progressively emerged as a valuable complementary approach, providing short- and long-term information into the assimilated diets of seabirds and allowing for the evaluation of their trophic niches [29,39,57]. Carbon stable isotope values ($\delta^{13}$C) can indicate the relative contribution of marine versus terrestrial food sources within gull diets; for instance, marine prey typically show higher $\delta^{13}$C values in comparison to terrestrial resources [40,57]. As such, $\delta^{13}$C values serve as a proxy for habitat use [58], allowing for the identification of foraging habitats and strategies [21,40]. Nitrogen stable isotope values ($\delta^{15}$N) are used as a proxy for estimating trophic levels [58], including the importance of fishery discards within the gulls' dietary compositions [29,59]. Additionally, SIA can be used to infer changes in the bird's diet during different or specific periods of the year through the analysis of tissues with different turnover rates [60]. Because blood components exhibit specific turnover rates, the analyses of plasma provides values referring to the diet from 1–2 weeks before sampling, i.e., incubation period, while red blood cells (RBC) provides information referring up to around 1–2 months prior to the sampling event, i.e., pre-laying period [60].

Blood samples of ca. 0.5 mL from the tarsal vein were collected from each sampled bird, using 27G needles, for isotopic analyses. The time taken for each bird from capture

to release was approximately 10 min. Up to 2 h after collection, the blood samples were separated into plasma and blood cells, using a centrifuge for 15 min at $1250 \times g$ and stored frozen ($-20\,°C$) until isotopic analyses.

$\delta^{15}N$ and $\delta^{13}C$ values for both blood plasma and cells of YLG were analysed, and a Bayesian dual-isotope mixing model was used to estimate the isotopic niches during the incubation and pre-laying periods for each year [61]. Lipids were removed from plasma by successive rinsing in a 2:1 chloroform/methanol solution to avoid the depletion of $\delta^{13}C$ values that can be induced by the high lipid content in plasma [62]. The low lipid content of RBC does not require delipidation [63]. The C/N mass ratio was $3.4 \pm 0.3$ (mean $\pm$ SD) in RBC and $3.7 \pm 0.2$ in plasma, below the 4.0 threshold, which corresponds to a low lipid concentration in the tissues [64].

A Flash EA1112 Series elemental analyser was used to determine carbon and nitrogen stable isotopic composition, coupled online via Finnigan conflo II interface to a Thermo Delta V S mass spectrometer. For the determination of nitrogen and carbon isotope ratios, c.a. 0.3 mg of each sample was combusted in a tin cup. The usual $\delta$ notation of isotope ratios are used, based on the Vienna PeeDee Belemnite (V-PDB) for carbon, and for nitrogen, atmospheric N2 (AIR), expressed as‰. $\delta^{13}C$ or $\delta^{15}N = [(Rsample/Rstandard) - 1] \times 1000$, where $R = {}^{13}C/{}^{12}C$ or ${}^{15}N/{}^{14}N$, respectively. A precision of <0.2‰ was indicated by replicate measurements of internal laboratory standards (acetanilide) for both $\delta^{13}C$ and $\delta^{15}N$.

*2.4. Oceanographic Parameters*

Three environmental parameters were used to evaluate oceanographic conditions around the breeding colony: Surface Chlorophyll-a concentration (Chl-a), Sea Surface Temperature (SST), and the extended winter North Atlantic Oscillation Index (wNAO). These parameters were chosen because they were found to potentially influence the availability of marine resources for seabirds in this area of the North Atlantic [40]. To characterise marine productivity, the dataset for Chl-a and SST spanning the interval from 2003 to 2022 were sourced from the Giovanni website https://giovanni.gsfc.nasa.gov/giovanni/ (accessed on 19 January 2023) for the months of March, April, May, and June. These months were chosen to coincide with the sampling period for Berlenga YLGs and their corresponding breeding period (i.e., May and June). Additionally, March and April were investigated to explore their potential impact on the initial clutches that are laid in late April, as well as their influence on the body reserves and fitness of adult YLGs, ultimately affecting their breeding success. Data prior to 2003 were not available, as the Aqua-MODIS satellite commenced operations only in July 2002. For both Chl-a and SST, the chosen plot type was "Time Series, Area-Averaged", and the coordinates (Bounding Box and Shape) were delineated at $9.57°$ W, $39.35°$ N, and $9.45°$ W, $39.46°$ N, with the area around the colony defining an area approximately $12 \times 11$ km in size around the colony. This area represents the core foraging grounds used by YLGs from Berlenga Island at sea [21,40]. A spatial resolution of 4 km was employed, with monthly composites derived from satellite-attained Chl-a and SST readings. For SST, the selected period was the night-time due to its lower amplitude of variation compared to diurnal SST.

The extended winter NAO index—December to March—was used because it has long-lasting effects on oceanic productivity patterns throughout each respective year [65], and its confirmed influence on YLG foraging strategies [40]. A year characterised by a negative wNAO presents low prey recruitment and availability for seabirds in this area of the North Atlantic due to an increase in warm conditions, low wind speed, and less vertical water mixing along the Iberian Peninsula, leading to a subsequent decline in marine productivity. Conversely, a positive phase of the wNAO is associated with a typical increase in marine productivity and the consequent availability of marine resources. However, exceptionally elevated values can potentially trigger strong upwelling events during the spawning season of small pelagic fish, displacing plankton and fish larvae away from the continental shelf, thereby compromising the availability of resources [66].

### 2.5. Fisheries Data Collection

Data on fisheries landings (traded at fish auction, in kg, by species, year, and month) were obtained from the main fishing harbour in the designated area, Peniche harbour (Figure 1), since 2002 (no available data for preceding years), as a proxy of the yearly fluctuations in fish availability within the study region, during the period of interest: March, April, May and June. The specific taxonomic groups of fish considered were the 10 most frequently consumed by the YLG on Berlenga Island [37]: the pelagic European pilchard (*Sardina pilchardus*), Atlantic mackerel and Atlantic chub mackerel (*Scomber* spp.), Atlantic horse mackerel, blue jack mackerel and Mediterranean horse mackerel (*Trachurus* spp.), garfish (*Belone belone*), and the demersal blue whiting (*Micromesistius poutassou*), bogue (*Boops boops*), European conger (*Conger conger*), zebra seabream, sheephead bream, white seabream and common two-banded seabream (*Diplodus* spp.), and pouting (*Trisopterus luscus*) and European hake (*Merluccius merluccius*). Fisheries landings were analysed by year during March–June, considering the total fisheries landings (March–June), fish taxonomic groups separately and by groups of 'pelagic' and 'demersal' fish species, and 'other species', which includes all the other species landed at Peniche harbour. Additionally, fisheries landings were also analysed by month (i.e., March, April, May, and June), and focusing on distinct fishing gear types: trawl, purse seine, and polyvalent fishing.

### 2.6. Urban Waste

Data on urban waste were used to evaluate their impact on the breeding performance (i.e., total eggs pair$^{-1}$, first egg, egg volume, and hatching success) and trophic niches of YLGs. The annual weight of residues delivered to the proximate active landfill (2010–2021, ValorSul, ca. 40 km Southeast of the breeding colony) was used as a proxy for the accessible food derived from the landfill, obtained from https://apambiente.pt/residuos/dados-sobre-residuos-urbanos (accessed on 14 February 2023, published by the Portuguese Environment Agency). Additionally, the annual weight of non-differentiated residues (2002 and 2009–2020) collected in the city of Peniche and for the broader West region, accessible at https://www.pordata.pt/Municipios/Res%c3%adduos+urbanos+total+e+por+tipo+de+recolha-655 (accessed on 14 February 2023), was used as a proxy for available food in urban settings.

### 2.7. Data Analysis

For the analysis of stable isotopic data, which aimed to build isotopic niches and perform comparisons between years, metrics based on a Bayesian framework SIBER: Stable Isotope Bayesian Ellipses in R (SIBER) [61] were employed. This methodology subjects the data to robust Bayesian statistical analyses, comprising three key metrics: total area (TA), standard ellipse area corrected for sample size (SEAc), and Bayesian standard ellipse area (SEA.B). TA refers to total convex hull area constructed from the extreme isotopic niche values, reflecting the total amount of niche space occupied, but it is very sensitive to sample size [67]. SEAc is the size of the standard ellipse obtained by Bayesian inference, accounting for sample size, and encompasses 40% of the data from the population of interest. SEA.B, the Bayesian model of the standard ellipse area, results from repeating the model a thousand times, enabling statistical comparisons of isotopic niche widths between different years. These metrics offer insights into resource utilisation patterns, indicating specialisation or generalisation in resource use. A larger isotopic niche width indicates greater consumption and exploitation of resources (prey and habitats). The package SIAR within R was used to compute the SIBER metrics. All analyses were performed using the package Commander under R 4.0.3 [68], and the significance level was set at $p < 0.05$.

Pearson's correlations were used to examine the relationships between the number of breeding individuals and wNAO. Linear models (LMs) were performed using the R package 'lme4' to evaluate the effect of the oceanographic conditions (wNAO, Chl-a, and SST), fisheries (species, months, and gear types), and urban waste (landfill, non-differentiated residues for Peniche, and the West region) on the number of breeding individuals and breeding performance (total eggs pair-1, hatching success, first egg, and egg volume).

Akaike's information criterion (AIC) and residual plots were utilised to identify the most suitable model fit for each response variable, considering the relevant explanatory variables for each conducted model. Significant explanatory variables were subsequently included.

The variables obtained through stable isotope analysis (SIA), which are indicative of trophic ecology, were employed in dual capacities as both roles as explanatory and response variables. Specifically, SIA variables ($\delta^{15}$N and $\delta^{13}$C values, TA, SEAc, and SEA.B) were considered as explanatory when evaluating their relationships with breeding performance. Conversely, in scenarios where the variables under analysis were oceanographic conditions, fisheries, and urban waste, SIA metrics were regarded as response variables. The most representative and integrative isotopic niche width metric, SEAc, was employed to examine relationships between trophic niche and oceanographic parameters, fisheries landings, and urban waste using linear models, while Pearson's correlations were used to assess specific relationships between isotopic metrics and the explanatory variables.

All data were tested for normality and homoscedasticity using Shapiro–Wilk and Levene's tests, respectively. The variable breeding individuals was square root transformed; Chl-a (May), fisheries (March–June), fisheries (April), fisheries (May), fisheries (June), pelagic species, purse seine, *Scomber* spp., *Trachurus* spp., *Micromesistius poutassou*, *Belone belone*, *Trisopterus luscus*, *Merluccius merluccius*, SEA.B (plasma), TA (RBC), SEAc (RBC), and SEA.B (RBC) were log10 transformed; and percentages were arcsine-transformed.

## 3. Results

### 3.1. Population Numbers (1974–2022)

The number of breeding YLGs on Berlenga Island can be separated into four distinct periods since 1974. The initial period (1974–1994) was characterised by an exponential increase in the number breeding individuals. The second period was brief (1994–1996) and characterised by an abrupt decline in population due to the culling of adult birds. Then, during the intermediate period (1997–2010), population numbers remained relatively stable with small inter-annual variations. From then on (2011–2022), there was a continuous decrease in the number of breeding individuals, and the population has not managed to recover its previous numbers (Figure 2).

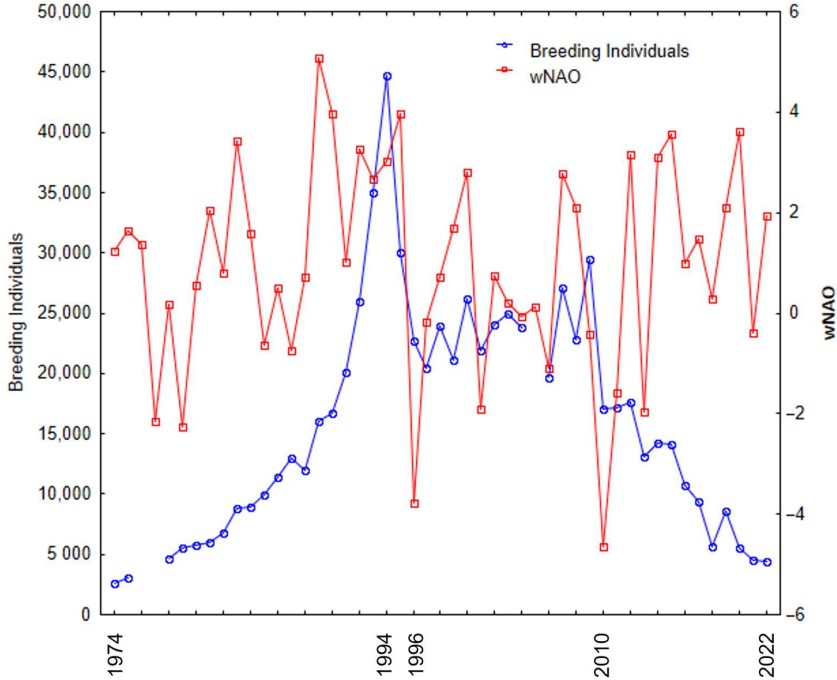

**Figure 2.** Number of breeding individuals in the Berlenga Island colony and wNAO values during the period from 1974 to 2022.

Overall, there was no significant relationship between the number of breeding individuals and wNAO during the period 1974–2022 (N = 46, r = 0.081, $p$ = 0.59; Figure 2). However, a notable parallel variation was evident between these two variables from 1997 to 2010 (N = 14, r = 0.622, $p$ = 0.023). In 2010, both the wNAO (a strongly negative value of −4.64) and the number of breeding individuals experienced a pronounced decline.

### 3.2. Relationships between Breeding Performance and Oceanographic Conditions (2002–2022)

A small decrease in Chl-a values and an elevation in SST were observed from March to June along the Portuguese coast, especially in the vicinity of Berlenga Island (Figure 3). However, no significant models were obtained when examining the relationship between environmental variables (wNAO, SST, Chl-a) and breeding performance metrics. Specifically, no significant associations were found for total eggs pair$^{-1}$ ($F_{2,17}$ = 1.2, adj $r^2$ = 0.01, $p$ = 0.30), first egg ($F_{1,16}$ = 2.4, adj $r^2$ = 0.08, $p$ = 0.14), egg volume ($F_{1,7}$ = 5.3, adj $r^2$ = 0.35, $p$ = 0.054), and hatching success ($F_{2,15}$ = 3.1, adj $r^2$ = 0.20, $p$ = 0.076).

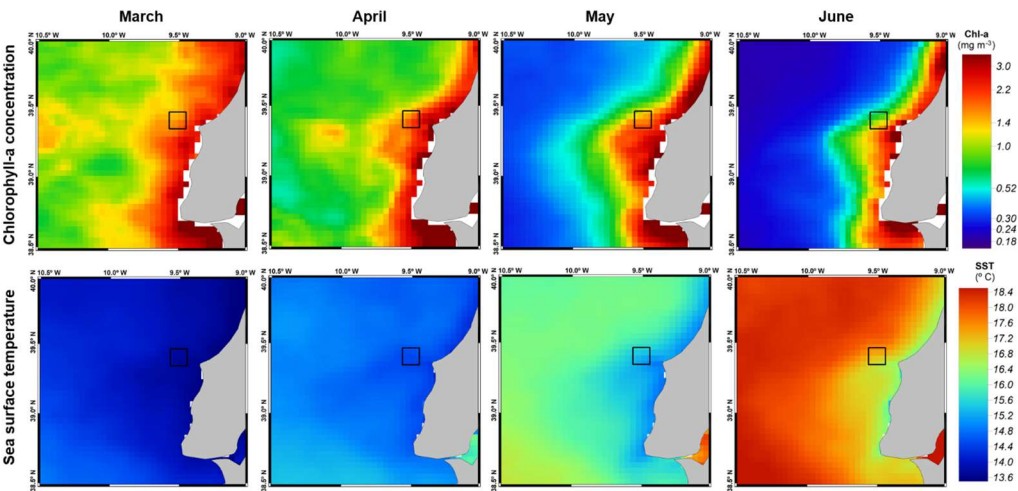

**Figure 3.** Climatological averages of Chl-a (**above**) and SST (**below**) for the months of March, April, May, and June, during the period 2002–2022. The black square indicates Berlengas Archipelago region, while the grey area corresponds to mainland Portugal.

### 3.3. Relationships between Breeding Performance and Fisheries Landings (2002–2022)

Several significant associations were observed between breeding performance and fisheries landings (categorised by species, and by month and gear). Significant models were identified between hatching success and fish species ($F_{1,17}$ = 4.6, adj $r^2$ = 0.17, $p$ = 0.047), revealing a positive correlation with *Trisopterus luscus* (t = 2.1, $p$ = 0.047). Additionally, our model confirmed that hatching success was significantly related to fishery landings ($F_{6,12}$ = 3.3, adj $r^2$ = 0.43, $p$ = 0.038), with positive associations found for the months of March (t = 2.5, $p$ = 0.027), April (t = 3.1, $p$ = 0.009), and May (t = 2.7, $p$ = 0.021), and negative associations with purse seine (t = −2.9, $p$ = 0.014) and trawl (t = −3.5, $p$ = 0.004) fisheries.

Furthermore, a significant model was identified between the number of eggs per breeding pair and fish species ($F_{5,14}$ = 4.4, adj $r^2$ = 0.47, $p$ = 0.013), with a positive relationship observed with *Micromesistius poutassou* (t = 3.4, $p$ = 0.005). However, no significant model was obtained between the number of eggs per breeding pair and fisheries landings categorised by month and gear ($F_{3,16}$ = 1.7, adj $r^2$ = 0.09, $p$ = 0.22).

The number of days to the first egg also show significant models with both fish species ($F_{3,15}$ = 10.6, adj $r^2$ = 0.62, $p$ < 0.001) and fisheries landings categorised by month and gear ($F_{2,16}$ = 4.7, adj $r^2$ = 0.29, $p$ = 0.025). Specifically, a positive relationship with *Sardina pilchardus* (t = 5.0, $p$ < 0.001) and a negative relationship with *Trisopterus luscus* (t = −3.6, $p$ = 0.003) were observed in relation to fish species, while a positive relationship with purse seine (t = 3.1, $p$ = 0.008) was found regarding fisheries landings.

In contrast, no significant relationships were detected between egg volume and data from fisheries landings whether categorised by fish species ($F_{1,7}$ = 2.0, adj $r^2$ = 0.11, $p$ = 0.20) or by month and gear ($F_{4,4}$ = 3.9, adj $r^2$ = 0.60, $p$ = 0.11).

### 3.4. Relationships between Breeding Performance and Urban Waste (2010–2021)

No significant relationships were found between the quantity of urban waste and hatching success ($F_{3,7}$ = 1.5, adj $r^2$ = 0.13, $p$ = 0.30), total eggs pair$^{-1}$ ($F_{3,7}$ = 0.5, adj $r^2$ = −0.18, $p$ = 0.69), or egg volume ($F_{2,4}$ = 0.2, adj $r^2$ = −0.35, $p$ = 0.81). Nevertheless, a significant model was obtained in relation to the number of days to the first egg and urban waste ($F_{1,11}$ = 5.0, adj $r^2$ = 0.25, $p$ = 0.047), specifically indicating a negative correlation with non-differentiated residues collected in the city of Peniche (t = −2.2, $p$ = 0.047).

### 3.5. Isotopic Values of YLG (2011–2022)

Among the years under investigation, 2013 exhibited the lowest mean $\delta^{13}$C values and the highest standard deviation (i.e., −20.6‰ ± 2.0) in plasma samples. In the remaining years, the mean $\delta^{13}$C values in plasma varied from −19.7‰ in 2022 to −18.4‰ in 2014 (Table 1). Regarding the mean $\delta^{15}$N values in plasma, these ranged from 12.0‰ in 2015 to 14.1‰ in 2021. In the RBC samples, the $\delta^{13}$C values ranged from −19.5‰ in 2016 to −18.4‰ in 2018, with 2013 again exhibiting the highest standard deviation (i.e., 1.3‰). The $\delta^{15}$N values of RBC ranged from 11.4‰ in 2012 to 13.8‰ in 2021 (Table 1).

**Table 1.** Stable isotope values ($\delta^{13}$C and $\delta^{15}$N) and isotopic niche metrics: the layman metric of convex hull area (TA), the area of the standard ellipse (SEAc, 40%), and the Bayesian approximation of the standard ellipse area (SEAb) in plasma and red blood cells (RBC) during the 2011–2022 period. Values are mean ± SD, with sample size in parenthesis.

| | 2011 (n = 26) | 2012 (n = 26) | 2013 (n = 6) | 2014 (n = 18) | 2015 (n = 9) | 2016 (n = 22) | 2017 (n = 16) | 2018 (n = 17) | 2021 (n = 15) | 2022 (n = 13) |
|---|---|---|---|---|---|---|---|---|---|---|
| $\delta^{13}$C (‰) Plasma | −18.6 ± 0.8 | −18.7 ± 0.5 | −20.6 ± 2.0 | −18.4 ± 0.4 | −18.8 ± 0.3 | −19.1 ± 0.7 | −19.3 ± 0.5 | −18.8 ± 0.7 | −19.2 ± 0.6 | −19.7 ± 0.6 |
| $\delta^{15}$N (‰) Plasma | 12.9 ± 1.5 | 12.2 ± 1.0 | 12.4 ± 1.9 | 13.4 ± 0.5 | 12.0 ± 0.9 | 12.7 ± 1.1 | 12.4 ± 0.8 (15) | 12.2 ± 1.2 | 14.1 ± 0.8 | 13.3 ± 0.5 |
| Plasma TA | 8.1 | 4.6 | 10.4 | 1.9 | 1.6 | 5.7 | 4.1 | 5.4 | 3.1 | 1.7 |
| Plasma SEAc | 3.0 | 1.5 | 13.1 | 0.7 | 1.0 | 2.5 | 1.5 | 2.7 | 1.3 | 1.1 |
| Plasma SEA.B | 3.1 ± 0.6 | 1.5 ± 0.3 | 13.0 ± 6.2 | 0.7 ± 0.2 | 1.0 ± 0.4 | 2.5 ± 0.6 | 1.5 ± 0.4 | 2.7 ± 0.7 | 1.3 ± 0.4 | 1.1 ± 0.3 |
| $\delta^{13}$C (‰) RBC | −19.3 ± 0.7 | −19.2 ± 0.6 | −19.1 ± 1.3 | −18.6 ± 0.6 | 18.5 ± 0.2 | −19.5 ± 0.7 | −18.8 ± 0.7 | −18.4 ± 0.3 | −19.4 ± 0.5 | −19.3 ± 0.6 |
| $\delta^{15}$N (‰) RBC | 12.7 ± 1.4 | 11.4 ± 1.1 | 11.6 ± 1.2 | 12.5 ± 0.9 | 11.7 ± 0.7 | 12.0 ± 1.2 | 12.6 ± 0.7 | 11.6 ± 0.9 | 13.8 ± 0.6 | 12.8 ± 0.7 |
| RBC TA | 7.3 | 4.7 | 5.2 | 2.9 | 0.6 | 7.8 | 2.8 | 1.9 | 2.8 | 2.7 |
| RBC SEAc | 2.2 | 1.4 | 6.3 | 1.1 | 0.4 | 2.7 | 1.4 | 0.8 | 1.1 | 1.3 |
| RBC SEA.B | 2.3 ± 0.5 | 1.5 ± 0.3 | 6.1 ± 3.0 | 1.2 ± 0.3 | 0.4 ± 0.2 | 2.7 ± 0.6 | 1.4 ± 0.4 | 0.8 ± 0.2 | 1.1 ± 0.3 | 1.3 ± 0.4 |

### 3.6. Isotopic Niches

The isotopic niche metrics (TA, SEAc, and SEA.B) calculated by SIBER analysis for both plasma and RBC showed that the highest overall values of isotopic niches corresponded to the years 2011, 2013 (notably), and 2016, while 2015 exhibited the lowest values (Table 1). Accordingly, the statistical comparison of ellipses size (SEA.B) indicated a significantly larger niche in 2013 compared to all other years, except for 2016, which was the second largest during the pre-laying period (i.e., based on RBC values) (Tables 2 and 3).

**Table 2.** Statistical comparisons between ellipses sizes representing isotopic niche widths using plasma. Significant *p*-values are indicated in bold.

| | 2011 | 2012 | 2013 | 2014 | 2015 | 2016 | 2017 | 2018 | 2021 | 2022 |
|---|---|---|---|---|---|---|---|---|---|---|
| 2011 | - | **0.006** | >0.999 | **0** | **0.003** | 0.204 | **0.009** | 0.298 | **0.005** | **0.001** |
| 2012 | | - | 1 | **0.011** | 0.104 | 0.947 | 0.429 | **0.961** | 0.321 | 0.130 |
| 2013 | | | - | **0** | **0** | **0** | **0** | **<0.001** | **0** | **0** |
| 2014 | | | | - | 0.717 | 1 | **0.979** | 1 | 0.960 | 0.843 |
| 2015 | | | | | - | 0.990 | 0.851 | **0.992** | 0.802 | 0.609 |
| 2016 | | | | | | - | 0.057 | 0.574 | **0.036** | **0.012** |
| 2017 | | | | | | | - | 0.949 | 0.393 | 0.182 |
| 2018 | | | | | | | | - | **0.032** | **0.007** |
| 2021 | | | | | | | | | - | 0.267 |
| 2022 | | | | | | | | | | - |

**Table 3.** Statistical comparisons between ellipses sizes representing isotopic niche widths using RBC. Significant *p*-values are indicated in bold.

|      | 2011 | 2012  | 2013  | 2014  | 2015   | 2016   | 2017   | 2018   | 2021   | 2022  |
|------|------|-------|-------|-------|--------|--------|--------|--------|--------|-------|
| 2011 | -    | 0.060 | **0.981** | **0.023** | **<0.001** | 0.718  | 0.080  | **<0.001** | **0.022** | 0.062 |
| 2012 |      | -     | **0.999** | 0.271 | **0.002**  | **0.979**  | 0.461  | **0.029**  | 0.219  | 0.375 |
| 2013 |      |       | -     | **0**     | **0**      | 0.054  | **<0.001** | **0**      | **<0.001** | **0.001** |
| 2014 |      |       |       | -     | **0.005**  | **0.994**  | 0.675  | 0.104  | 0.415  | 0.590 |
| 2015 |      |       |       |       | -      | **>0.999** | **0.997**  | **0.947**  | **0.986**  | **0.995** |
| 2016 |      |       |       |       |        | -      | **0.028**  | **0**      | **0.005**  | **0.025** |
| 2017 |      |       |       |       |        |        | -      | **0.050**  | 0.262  | 0.425 |
| 2018 |      |       |       |       |        |        |        | -      | 0.840  | 0.921 |
| 2021 |      |       |       |       |        |        |        |        | -      | 0.663 |
| 2022 |      |       |       |       |        |        |        |        |        | -     |

Isotopic niches represented by the ellipses clearly illustrate these differences in both $\delta^{13}C$ and $\delta^{15}N$ values for plasma and RBC. The larger isotopic niches of the years 2011, 2013 and 2016 contrast markedly with the considerably narrower isotopic niche width observed in 2015, indicating that YLGs exhibited a higher generalist degree in 2011, 2013, and 2016, while displaying greater specialisation in the resources exploited in 2015 (Figure 4).

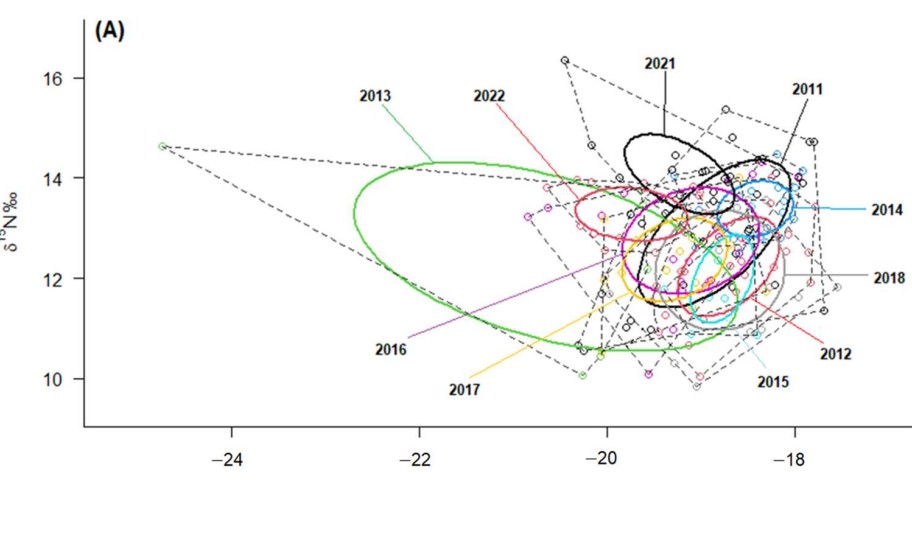

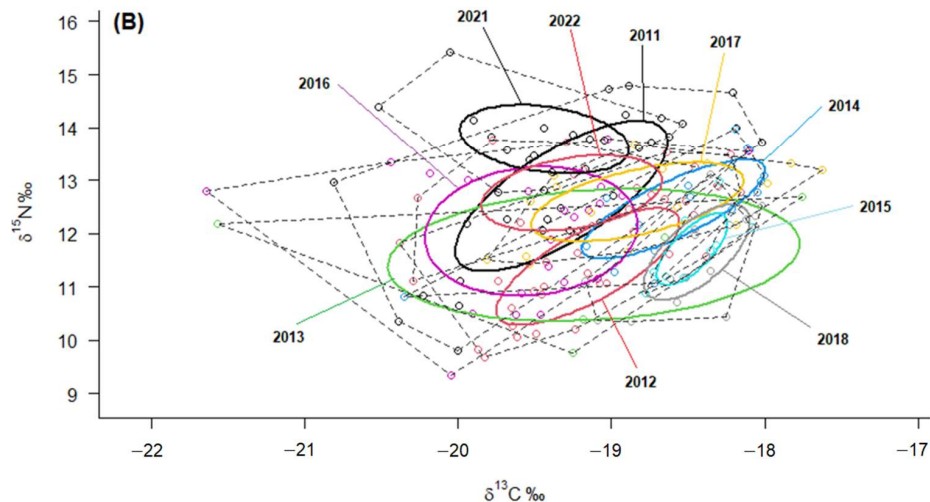

**Figure 4.** Stable isotope ratios in (**A**) plasma and (**B**) RBC of YLG breeding on Berlenga Island, from 2011 to 2022 (no data available for 2019 and 2020). The area of the standard ellipses (SEAc) is represented by solid lines and the total area of the convex hull (TA) is represented by dashed limits. Each colour represents a year, and each point represents a sampled individual.

*3.7. Relationships between Breeding Performance and Trophic Ecology (2011–2022)*

No significant models were identified when investigating the relationship between breeding performance and isotopic values measured from plasma ($\delta^{15}$N and $\delta^{13}$C values, TA, SEAc, and SEA.B). Specifically, no significant associations were found for hatching success ($F_{1,8}$ = 0.0, adj $r^2$ = −0.12, $p$ = 0.89), total eggs pair$^{-1}$ ($F_{1,8}$ = 4.3, adj $r^2$ = 0.27, $p$ = 0.071), or first egg ($F_{1,8}$ = 1.4, adj $r^2$ = 0.04, $p$ = 0.28).

On the other hand, a significant model was identified between breeding performance and isotopic values measured from RBC, particularly for hatching success ($F_{5,4}$ = 9.7, adj $r^2$ = 0.83, $p$ = 0.023). This model revealed negative relationships with $\delta^{15}$N (t = −3.0, $p$ = 0.039) and $\delta^{13}$C (t = −4.7, $p$ = 0.009) values. However, no significant models were found for total eggs pair$^{-1}$ ($F_{4,5}$ = 3.0, adj $r^2$ = 0.48, $p$ = 0.13) or the timing of the first egg ($F_{1,8}$ = 4.4, adj $r^2$ = 0.28, $p$ = 0.068).

*3.8. Relationships between Trophic Ecology with Oceanographic Parameters, Fisheries Landings, and Urban Waste (2011–2022)*

In general, isotopic signatures, as response variables, showed robust associations with the oceanographic parameters (i.e., wNAO, Chl-a, and SST). We highlight the negative significant correlations observed between wNAO and isotopic metrics measured via plasma (i.e., during incubation): SEAc (N = 10, r = −0.80, $p$ = 0.006), TA (N = 10, r = −0.81, $p$ = 0.004), and SEA.B (N = 10, r = −0.80, $p$ = 0.005). Similarly, negative correlations between wNAO and the isotopic metrics reflecting the pre-laying period were also detected: SEAc (N = 10, r = −0.67, $p$ = 0.036) and SEA.B (N = 10, r = −0.66, $p$ = 0.040).

In addition, March SST, April SST, and June SST exhibited positive correlations with RBC TA (N = 10, r = 0.65, $p$ = 0.043), RBC $\delta^{15}$N values (N = 10, r = 0.68, $p$ = 0.031) and plasma $\delta^{13}$C values (N = 10, r = 0.66, $p$ = 0.037), respectively. Positive significant correlations were also detected between Chl-a (May) and RBC $\delta^{13}$C values (N = 10, r = 0.68, $p$ = 0.029).

Linear models revealed significant relationships involving SEAc values derived from plasma ($F_{7,2}$ = 21.9, adj $r^2$ = 0.94, $p$ = 0.044). These relationships indicated positive associations with March SST (t = 7.4, $p$ = 0.018) and April SST (t = 7.2, $p$ = 0.019), as well as negative associations with May SST (t = −6.0, $p$ = 0.027), Chl-a (March) (t = −8.2, $p$ = 0.014), Chl-a (April) (t = −4.3, $p$ = 0.049), and Chl-a (March–June) (t = −5.1, $p$ = 0.036). Additionally, SEAc values obtained from RBC exhibited a significant relationship ($F_{1,8}$ = 5.4, adj $r^2$ = 0.33, $p$ = 0.049), specifically indicating a positive association with Chl-a (April) (t = 2.3, $p$ = 0.049).

There were also significant relationships between SIA variables, particularly $\delta^{15}$N values, and fisheries landings. Specifically, plasma $\delta^{15}$N values were positively correlated with *Diplodus* spp. (N = 10, r = 0.67, $p$ = 0.036), *Merluccius merluccius* (N = 10, r = 0.89, $p$ < 0.001), demersal fisheries (N = 10, r = 0.69, $p$ = 0.027), and fisheries landings (April) (N = 10, r = 0.71, $p$ = 0.021). Similarly, RBC $\delta^{15}$N values showed positive significant relationships with *Diplodus* spp. (N = 10, r = 0.80, $p$ = 0.005), *Merluccius merluccius* (N = 10, r = 0.82, $p$ = 0.004), pelagic species (N = 10, r = 0.71, $p$ = 0.021), purse seine (N = 10, r = 0.70, $p$ = 0.024), fisheries landings (March–June) (N = 10, r = 0.74, $p$ = 0.015) and fisheries landings (April) (N = 10, r = 0.80, $p$ = 0.005). Plasma TA also showed a positive significant relationship with other species landed (N = 10, r = 0.65, $p$ = 0.041). However, no significant linear models were obtained when accessing the relationship between SEAc values derived from plasma and fish landings categorised by species ($F_{7,2}$ = 10.9, adj $r^2$ = 0.89, $p$ = 0.087), or between SEAc values obtained from RBC and fish landings by species ($F_{4,5}$ = 2.3, adj $r^2$ = 0.36, $p$ = 0.20).

No significant relationships were detected between SIA variables and urban waste, and no significant linear models were identified when accessing the relationship between urban waste with SEAc values derived from plasma ($F_{1,6}$ = 0.2, adj $r^2$ = −0.13, $p$ = 0.68) or RBC ($F_{2,5}$ = 0.5, adj $r^2$ = −0.19, $p$ = 0.66).

## 4. Discussion

This study investigated the potential influences of oceanographic and anthropogenic factors, including population control measures, on the fluctuations in breeding measures of

YLGs. The results confirmed the influence of oceanographic conditions and, consequently, of marine productivity around Berlenga Island, on the trophic ecology of YLG. Overall, oceanographic conditions did not appear to significantly affect the breeding performance of this population on a larger scale, confirming its resilience, and that the population control measures seemed to have played a critical role in stabilising and even reducing the population. Results also suggested that sporadic periods of poor oceanographic productivity over the last decade influenced a faster and consistent reduction in the number of breeding individuals, intensified by the enduring effects of the population control measures.

The North Atlantic Oscillation index (NAO) has a broad influence on the marine ecosystems of the North Atlantic. Throughout the 1974–2022 period, there was no consistent correlation between the number of breeding individuals and wNAO. This variability can be attributed to the population dynamics, including an exponential increase in the number of breeding individuals until 1994, followed by a sudden decline due to the culling of adult birds during 1994–1996. However, from 1997 until the year 2010, when population numbers relatively stabilised, these two variables displayed synchronous oscillations and a significant positive relationship was evident. Subsequently, a continuous declining trend was observed until 2022. In 2010, an extraordinarily low wNAO value of −4.64 was recorded, impacting the ecosystem by causing a drastic reduction in marine productivity, and affecting higher trophic level consumers, such as seabirds [46]. The observed data suggest that 2010 marked a tipping point for the gull on Berlenga Island, but current estimates for 2022 still suggest around 4300 breeding individuals in this colony [36]. Egg culling has been employed as a control measure since 1999, which seems to have resulted in a current population consisting primarily of individuals aged approximately 23 years and older. Furthermore, older individuals may have experienced a reduction in the quality of their eggs, diminished capacity to nourish their offspring, and a decline in their ability to defend both their territory and their young. This demographic shift partially accounted for the decline in breeding adults observed since 2010. Essentially, the older individuals reached the end of their lifespans, and few individuals entered the reproductive stage. The observed decrease in the density of nests in control areas suggested that there was no influx of breeding individuals. Even if there was an influx of migrants from other colonies [35], it would not have been sufficient to compensate for the number of breeding individuals that had perished [36].

In contrast to our initial hypothesis, which suggested that reproductive performance would be affected by the quality of oceanographic conditions, the absence of significant relationships between oceanographic conditions (wNAO, SST, and Chl-a) and the breeding measures of YLGs on Berlenga Island, indicated that oceanographic conditions did not exert a significant impact on the breeding performance of YLGs on a larger scale. However, our investigation revealed that oceanographic conditions did have an influence on the trophic niche of YLGs. Several connections were observed between trophic ecology (i.e., SIA variables) and environmental variables (detailed below). There is evidence that gulls tend to rely more heavily on refuse dumps when marine resources become scarcer and oceanographic conditions become poorer [21,40]. On the other hand, the diet of adults during the breeding season may comprise a considerable proportion of natural sources during favourable oceanographic conditions, such as the Henslow's swimming crab *Polybius henslowii* when available [21,37]. Therefore, substantial temporal variations in oceanographic conditions determining the abundance of marine resources might occur, such as those indicated by the Chl-a concentration, which correlates positively with the density of low-trophic level marine species like the Henslow's swimming crab [69]. These fluctuations can significantly impact the trophic ecology of both mid-trophic level species [70] and top predators like seabirds [71]. However, direct observed effects on breeding performance may be more challenging to discern. Such shifts in resource availability are consistent across different regions, years, and even during distinct phases of the breeding season [21,37,38], and are closely linked to shifts in oceanographic conditions, particularly in relation to NAO values [4,40]. A study conducted on a colony of little terns *Sternula albifrons* in Ria Formosa,

South Portugal, demonstrated that negative NAO values were associated with earlier breeding onset, and lower SSTs during the winter–spring period resulted in increased prey fish abundance, subsequently influencing their reproductive parameters [4].

It is already known that YLG from the Berlenga Island exploit the Peniche harbour to feed on fisheries leftovers [21], and that during the breeding season, they tend to rely more on marine resources [38,40]. While one might expect that a higher resource availability would correspond to elevated egg productivity and larger population numbers [27], this inference for Berlenga cannot be drawn due the potential biases resulted from the implementation of control measures. YLG lay an average of 2–3 eggs per clutch [72], but the eggs on Berlenga Island are subjected to culling (with the exception of the control areas where no culling measures are taken), leading some pairs to continue replenishing their clutches with up to two and rarely three sets of eggs. In several bird species, a decrease in egg size is observed in the last laid eggs. A study on the YLG colony of Comacchio lagoon, NE Italy, showed that the mass of the third egg is enhanced due to increased albumen content, indicating that when ecological conditions are favourable, the third egg has greater chances of survival, and thus the entire clutch [73]. Consequently, the subsequent clutches of YLG on Berlenga would be potentially smaller in both number of eggs and egg size, especially when oceanographic conditions are poorer. This situation could influence the survival prospects of hatchlings from eggs that survive culling. When combined with hatching asynchrony, this phenomenon results in size disparities among siblings and clutches. Consequently, during competition for food, it could lead to the death of smaller and weaker individuals [74,75].

Several relationships between breeding performance (from the control areas, where clutches are not culled) and fisheries landings were identified. Specifically, significant models revealed positive correlations between hatching success and fish species, particularly with *T. luscus*, and between hatching success and fishery landings, with positive associations identified for the months of March, April, and May, while negative associations were found with purse seine and trawl fisheries. Additionally, a significant model was obtained between the total number of eggs per breeding pair and fish species, exhibiting a positive relationship, particularly with *M. poutassou*. The number of days to the first egg also showed a significant relationship with fish species, with a positive relationship observed with *S. pilchardus* and a negative relationship with *T. luscus*. On the other hand, the time taken to lay the first egg showed a negative correlation with the decrease in the amount of urban waste for the city of Peniche. These results indicated that during periods of high fisheries activity (especially from purse seine targeting pelagic species such as *Sardina pilchardus*), YLGs exhibit delayed clutch initiation, likely to mitigate potential competition among conspecifics and improve body reserves. In contrast, in years with reduced fishing activity, gulls engaged in greater scavenging from urban waste sources, and began laying earlier. One pivotal species of both ecological and economic significance is the sardine *S. pilchardus*, which is prominently caught using purse seine, and one of YLG's most important prey [29,37]. However, the reduction in landings of *S. pilchardus* since 2005 has resulted in the importance of other significant species in the YLG's diet, including fish living at deeper depths, such as *T. luscus* and *M. poutassou*, that should be made available to gulls through fisheries discards [37,38]. However, these substitutions might not adequately meet the nutritional requirements of YLG that primarily rely on *S. pilchardus* [76].

Human activities, such as fisheries and urban waste management, exert influence on the diet of gulls, potentially inducing alterations in trophic niches depending on the resources consumed [77]. In our study both $\delta^{13}$C and $\delta^{15}$N values showed a large variation over the years, in both RBC (representing the pre-laying) and plasma (reflecting incubation). We identified significant relationships between $\delta^{15}$N values and fisheries landings, particularly in the case of demersal carnivorous species (e.g., *Diplodus* spp., *Merluccius merluccius*), which are more likely to exhibit higher $\delta^{15}$N values [29,70]. These species are unlikely to be captured by gulls under natural conditions, further supporting the idea of a fisheries subsidy, particularly through discards, to their diet [37]. Despite the small

sample size, 2013 showed more negative $\delta^{13}$C values compared to other years, which should indicate the increased consumption of terrestrial-origin sources by the gulls in 2013 [40]. This is also supported by the fact that isotopic niches from both tissues were larger and had a high variation in 2013. Investigations involving stable isotopes and pellets from a colony of Audouin's gulls *Ichthyaetus audouinii* in Ria Formosa (South Portugal) in 2013, revealed reduced trophic levels in chicks compared to other years [29]. This was attributed to unfavourable oceanographic conditions, potentially driving adult gulls to intensify foraging efforts in offshore areas, concurrent with an unusually low wNAO value of −1.97. Furthermore, the years 2011, 2013, and 2016 presented the highest isotopic niche values for both plasma and RBC (accompanied by low wNAO values), whereas 2015 showcased the lowest isotopic niche values (linked to a high wNAO value). In fact, isotopic signatures showed strong correlations with oceanographic parameters (wNAO, Chl-a, and SST). Our analyses revealed that wNAO had negative correlations with the isotopic niche metrics, representing the overall breadth of the YLG population's dietary niche during the pre-laying and incubation periods. This suggests that higher wNAO values drive narrower niche widths and heightened specialisation in marine resources. On the contrary, in years with negative wNAO, the dietary niche expanded, and gulls adopted a more generalist feeding involving a higher mixture of resources from both terrestrial and marine origins. This adaptability enables gulls to rapidly modify their behavioural patterns to exploit available resources, and the consequences of such adjustments, including trophic shifts, are evident at the individual and populational levels [15,21,78]. In addition, several relationships were detected between SIA variables with both SST and Chl-a, including models involving SEAc values, highlighting the influence of oceanographic dynamics on the trophic ecology of gulls. Such isotopic composition differences are commonly linked to environmental drivers, such as SST and Chl-a concentration in species spanning from mid- to higher trophic levels within the study area [70,71]. These findings underline the significant influence of oceanographic conditions on the trophic ecology of gulls. However, interpreting specific and indirect associations with their trophic ecology is challenging, and drawing definitive conclusions is difficult. This challenge arises because birds do not directly feed on Chl-a concentration and can scavenge from a wide variety of resources. Nevertheless, these conditions did not appear to have direct repercussions on breeding performance in our study population (up to egg-hatching), highlighting the notable resilience of this gull species.

## 5. Conclusions

This study offers valuable insights into the dynamics of the yellow-legged gull (YLG) colony on Berlenga Island, the largest in Portugal, with a monitoring history dating back to 1974. By analysing population data and stable isotopic compositions, this research revealed complex connections between environmental factors, population dynamics, breeding success, and trophic interactions.

Stable isotope analysis broadened insights into the gulls' trophic behaviour, with isotopic signatures reflecting shifts in oceanographic parameters and resource availability. Contrary to our expectations, direct correlations between oceanographic variables and breeding performance were not found, possibly confounded by effective culling population control. However, the influence of marine productivity fluctuations driven by parameters like wNAO and Chl-a concentration, and SST was evident in the isotopic signatures of YLGs. These dynamics influenced their trophic ecology and suggested potential cascading effects on reproductive outcomes. As expected, the gulls displayed adaptability in their foraging behaviour, emphasising terrestrial sources during periods of marine resource scarcity and unfavourable oceanographic conditions and broadening their trophic niches. The models conducted unveiled significant relationships between reproductive performance and fisheries landings, especially concerning demersal species that are expected to be accessible to gulls through fisheries but unlikely to be captured by gulls under natural conditions.

Despite the recent decline in YLG population size, the necessity for population control measures, specifically egg culling, remains. While the population has decreased, the current number of breeding individuals remains considerably higher than the numbers recorded in the 1970s, emphasising the ongoing need for these interventions to maintain sustainable population levels in this delicate ecosystem (UNESCO Biosphere Reserve). However, it should be emphasised that the root cause of this overpopulation is situated upstream, specifically due to the gulls' access to man-made food sources. Thus, for efficient control, not only in Berlengas Reserve but also in the city of Peniche, the issue of urban waste and fisheries leftovers should be carefully managed, given the resilient nature of this opportunistic species. As future cohorts persist in their occupation of the island, these insights will serve as a base for informed conservation strategies. It is essential to consider the relationship between natural and anthropogenic factors in shaping the trajectory of YLG populations.

**Author Contributions:** Conceptualisation, F.R.C., N.C.S., E.A.S. and J.A.R.; formal analysis, F.R.C. and N.C.S.; funding acquisition, F.R.C., V.H.P., L.M. and J.A.R.; investigation, F.R.C., N.C.S. and J.A.R.; methodology, F.R.C., N.C.S., V.H.P., L.M. and J.A.R.; resources, F.R.C., V.H.P., L.M. and J.A.R.; visualisation, F.R.C., N.C.S. and J.A.R.; writing—original draft, F.R.C. and N.C.S.; writing—review and editing, F.R.C., N.C.S., V.H.P., L.M., E.A.S. and J.A.R. All authors have read and agreed to the published version of the manuscript.

**Funding:** This study had the support of national funds through Fundação para a Ciência e Tecnologia, I. P (FCT; Portugal), under the projects UIDB/04292/2020 and UIDP/04292/2020 granted to MARE, LA/P/0069/2020 granted to the Associate Laboratory ARNET, and the transitory norm contract DL57/2016 granted to Filipe R. Ceia (DL57/2016/CP1370/CT90) at the University of Coimbra. The funding sources had no involvement in this study.

**Institutional Review Board Statement:** The animal study protocol was approved by the Institutional Review Board of the Portuguese Government (Institute for Nature Conservation and Forests—ICNF) under licenses: 89/2011/CAPT (2011), 24/2012/CAPT (2012), 42/2013/CAPT (2013), 57/2014/CAPT (2014), 18/2015/CAPT (2015), 69/2016/CAPT (2016), 101/2017/CAPT (2017), 98/2018/CAPT (2018), 517/2021/CAPT (2021), and 143/2022/TRANSP (2022).

**Data Availability Statement:** The datasets used and/or analysed during the current study are available from the corresponding author on reasonable request.

**Acknowledgments:** We are thankful for the fishery landings data supplied by Dados Estatísticos (2022), Docapesca—Portos e Lotas, S.A. The authors would like to thank the Institute for Nature Conservation and Forests (ICNF) and Berlengas Nature Reserve for permissions to work on Berlenga Island. Special thanks to wardens Paulo Crisóstomo, Eduardo Mourato, Alexandre Bouça, Sérgio Borges, Márcio Duarte, Ana Santos, Nuno Dias, Tiago Menino, Filipe Correia, and many other wardens from other protected areas to undertake on the population control measures implemented at Berlenga Island and collected data on breeding population. We also thank all the students and collaborators who participated in the field.

**Conflicts of Interest:** The authors declare no conflict of interest.

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
