# Peer review of "Gulls as Indicators of Environmental Changes in the North Atlantic: A Long-Term Study on Berlenga Island, Western Portugal"

_diversity, doi:10.3390/d15111148_

Round 1

Reviewer 1 Report

Comments and Suggestions for Authors

While the authors have collected an impressive amount of data, the methods need a lot more details and clarification and I think there are issues in the way the data were analysed making the interpretation of the results difficult to assess.

MAJOR COMMENTS

Abstract

In general, the abstract is too vague and lacks details. For example, you mention that some control measures were put in place (l16) but do not mention what those measures were. You also mention there were some strong relationships between trophic ecology, oceanographic parameters, and fisheries landings but you do not tell us the direction(s) of the relationship(s) or which aspects of tropic ecology were used in the analyses. Which oceanographic parameters did you measure? What do you call fisheries landings? Etc… I also strongly suggest using active voice rather than passive voice here and throughout the manuscript (e.g., a long-term database was related with... reads awkwardly).

Introduction

It is generally best to start your first paragraph with the bigger picture? The broad scale question that you are asking in the paper and that is relevant to the general readers. What is the big knowledge gap that this study will fill? I can see some of this information in the second paragraph, but I suggest bringing this up front.

The introduction is also very focused on the study species and should be broadened to be relevant to a wider audience.

Methods

L155. How were the census conducted? By how many individuals? At what time of the year? Manually (i.e., walking within the colony)? More details are needed.

L159 and Figure 1. Do the bird breed all over the island or only along the coast? Currently it is hard to assess how much of the population the two control areas encompass. How were these areas chosen? If all the active nests were gps marked, you could add the gps points of the nests to give a better representation of the spread of the birds over the island.

L159. Again, more details are needed. How is egg culling conducted? Does this occur for every single egg encountered (expect in the control areas)? Etc…

L168. Some information on the breeding ecology of the birds is needed. When do the bird breed? Are they asynchronous or synchronous? What is the average clutch size? How many clutches do a pair produce per season? Etc…

L222-250. Better justification needs to be given for the chosen months and areas. Why only two months prior to breeding? Why December to March for the NAO index and not December to June? For example. How much of the marine area sampled overlapped with the foraging of the gulls? Please add the distance in km from the coast this encompassed and the justification for this selection. Why did you focus on SST and Chl? Winds can also be an important parameter for breeding and foraging success in seabirds for example.

L272-280. Why not on a monthly basis? Where is the landfill in relation to the breeding colony? What do you mean 3 potential food sources?

L281.314. The stats section needs clarity and needs to be completely re-written. Currently, this is impossible to determine what has been done. Please refer back to your aims and ensure that the analyses are presented in the order of the aims. At the moment, we have to fish for information in every sections. I also don’t think that Pearson’s correlations are correct. You have yearly and monthly effects, so you need to consider time series analyses. What is SIA and which variables were obtained form this? Limit abbreviations as much as possible as this makes it difficult to remember them all.

Results

You need to report the statistical results even for non-significant relationships or refer to tables presenting the statistical results. Following on my comments regarding the statistical analyses, it also appears like you have run separate correlations while you need to combine all factors into the same analysis. I have not commented further on the results as the statistics need to be completely re-done.

The sample size for each steps need to be clearly presented. At the moment, it is unclear how many individuals you sampled from and thus the representativeness of the results. From the methods, it appears as if you only sampled individuals in the control zone. Is this correct? This would represent only a small number of individuals (although, I’m only guessing here – see my comment above re: L159). How did you ensure that you did not resample the same individuals across years? Same with the number of eggs measured? How many eggs were measured and from how many pairs?

Discussion

The discussion will need to be re-written once the results have been re-analysed. In general, however, the discussion is too narrowly focused on the study species and needs to be broadened.

MINOR COMMENTS

Abstract

L15. “There was exponential growth of the population until 1994” should be changed to “The population grew exponentially”.

L26. “…at a faster rate”. Faster than what?

Introduction

L33. What impacts?

L44-47. Ok but what were the results of these investigations?

L52-54, L74-75, L85-86. You need some references here.

L67. How much of the population did this represent?

L79. In which way?

L126. Please specify which specific parameters, data and records were obtained. What do you mean by fisheries landings (L132)?

L136-138. Unnecessary. I would replace however with some predictions.

Methods

L142. Data is always plural. Please correct here and through the manuscript. Which specific data? The bird data or all data?

L167-169. Why is this information important? Re-state.

L173-176. This information should be moved to 2.3

L177-181. This information should be moved to 2.4.

L183. But you said above that no bird were captured in 2019.

L185-187. Please break this sentence into smaller sentences. Statistics and variables used for the analyses should be clearly stated and moved to the stats section.

L204-220. If you are using standard methods for this analysis, you should add some references.

L152. What do you mean by fisheries landings? Number of fishes caught?

L264-265. Specify which species were allocated to which group.

L266-227. Unclear.

Discussion

Why do you switch to present tense in the discussion?

What is the return rate in the population? Do you have any information on survival rate and recruitment? This should be discussed or at least acknowledged.

Comments on the Quality of English Language

Few issues as noted in the Minor Comments. 

Author Response

Thank you for reviewing our manuscript and for suggesting improvements. We are grateful for the careful reviews and a revision of the ms. was performed as recommended. We have revised the respective parts of the ms. following the helpful comments, particularly in the introduction and methods sections. The text has been improved, and we conducted a re-analysis using linear models (LMs) to ensure robust results, as suggested. Taking into consideration the limited number of up to 200 words, the abstract was also improved in line with the provided comments. We are now submitting a revision of the manuscript addressing, point-by-point, the issues raised in the comments.

Comments and Suggestions for Authors

While the authors have collected an impressive amount of data, the methods need a lot more details and clarification and I think there are issues in the way the data were analysed making the interpretation of the results difficult to assess.

Reply: Thank you for your feedback and for taking the time to review our work. The methods are now more detailed and clarified. We have made the necessary revisions to offer a comprehensive explanation of our data collection and analysis methods. Additionally, we conducted new analyses to ensure the robustness of our results.

MAJOR COMMENTS

Abstract

In general, the abstract is too vague and lacks details. For example, you mention that some control measures were put in place (l16) but do not mention what those measures were. You also mention there were some strong relationships between trophic ecology, oceanographic parameters, and fisheries landings but you do not tell us the direction(s) of the relationship(s) or which aspects of tropic ecology were used in the analyses. Which oceanographic parameters did you measure? What do you call fisheries landings? Etc… I also strongly suggest using active voice rather than passive voice here and throughout the manuscript (e.g., a long-term database was related with... reads awkwardly).

Reply: The abstract has been rewritten to offer a more detailed and clear overview of the study, and it has been enhanced in accordance with the new analyses conducted. We understand the challenge of condensing substantial information into a 200-word paragraph, but we have addressed your concerns. The control measures have been clearly identified, the relationships described in more detail, and specific oceanographic parameters and the definition of fisheries landings have been provided for better clarity.

Introduction

It is generally best to start your first paragraph with the bigger picture? The broad scale question that you are asking in the paper and that is relevant to the general readers. What is the big knowledge gap that this study will fill? I can see some of this information in the second paragraph, but I suggest bringing this up front.

Reply: We accept the recommendation of the reviewer 1 and we have restructured the introduction to commence with the bigger picture, emphasizing the broad-scale question of significance to general readers. The information originally found in the second paragraph is now presented upfront in the introduction, as suggested.

The introduction is also very focused on the study species and should be broadened to be relevant to a wider audience.

Reply: Additional information has been incorporated into the introduction (e.g., L39-42; L52-57; L67-71) to broaden the relevance of this study to a wider audience.

Methods

L155. How were the census conducted? By how many individuals? At what time of the year? Manually (i.e., walking within the colony)? More details are needed.

Reply: We have provided additional details and included supporting references on how the census was conducted (L182-186) ‘The census of breeding birds is conducted by two ICNF wardens and starts approximately 1.5 hours before sunset, during two consecutive days, at which point the majority of birds have already returned to the colony. On one day, the birds present on the slopes of Berlenga are counted, and on the other day, those on the plateau [56]. The census takes place at the beginning of the incubation period (i.e., early May).’

L159 and Figure 1. Do the bird breed all over the island or only along the coast? Currently it is hard to assess how much of the population the two control areas encompass. How were these areas chosen? If all the active nests were gps marked, you could add the gps points of the nests to give a better representation of the spread of the birds over the island.

Reply: We have provided additional details on egg culling campaigns to clarify this methodology (L187-195) ‘Egg culling is carried out on Berlenga Island since 1999, and campaigns are scheduled based on the timing of reproduction (egg laying - approximately late April to mid-June; incubation - around 28 days), the size of the colony, maximization of efficiency, and minimizing costs. The participants (ICNF wardens) have prior experience to avoid missing nests. Seven consecutive days are required for the entire island to be thoroughly surveyed with a team of 8 wardens. The entire island is traversed on foot, with the exception of inaccessible cliffs and two designated permanent control areas, each spanning 800 m². The two control areas are visited every 3 days, one located on the slope and the other on the plateau (Figure 1), for the assessment of annual breeding success [36]’. The birds were sampled throughout the island (N=168), mainly on the plateau but also on the slopes of the island over the 2011-2022 period.

L159. Again, more details are needed. How is egg culling conducted? Does this occur for every single egg encountered (expect in the control areas)? Etc…

Reply: We now provide more details on how egg culling is conducted by providing additional information and including supporting references in the relevant section.

L168. Some information on the breeding ecology of the birds is needed. When do the bird breed? Are they asynchronous or synchronous? What is the average clutch size? How many clutches do a pair produce per season? Etc…

Reply: New information regarding the breeding ecology of the bird has been added to the methods (L187-190) ‘Egg culling is carried out on Berlenga Island since 1999, and campaigns are scheduled based on the timing of reproduction (egg laying - approximately late April to mid-June; incubation - around 28 days), the size of the colony, maximization of efficiency, and minimizing costs.’ Additionally, details about the asynchronous nature of the bird's breeding and the handling of their clutches have been included in L195-200 ‘Given the nature of YLGs replacing their clutches and the asynchronous nature of the initial egg laying, ICNF wardens undertake three separate egg-culling campaigns. The three campaigns are spaced at 21-day intervals (between May and June), to ensure that no new gull chicks hatch in the meantime and to allow gulls, who had their eggs culled, to replace them (approximately 30% of gull pairs can replace their eggs once or twice).’

 L222-250. Better justification needs to be given for the chosen months and areas. Why only two months prior to breeding? Why December to March for the NAO index and not December to June? For example. How much of the marine area sampled overlapped with the foraging of the gulls? Please add the distance in km from the coast this encompassed and the justification for this selection. Why did you focus on SST and Chl? Winds can also be an important parameter for breeding and foraging success in seabirds for example.

Reply: All this information is now clarified and supported with appropriate references. This information is now clarified in the response as follows:

For the choice of parameters (L262-263) ‘These parameters were chosen because they were found to potentially influence the availability of marine resources for seabirds in this area of the North Atlantic [40,66].’

For the selection of months (L266-270) ‘These months were chosen to coincide with the sampling period for Berlenga YLGs and their corresponding breeding period (i.e., May and June). Additionally, March and April were investigated to explore their potential impact on the initial clutches that are laid in late April, as well as their influence on the body reserves and fitness of adult YLGs, ultimately affecting their breeding success.’

For the selection of the marine area (L272-276) ‘For both Chl-a and SST the chosen plot type was “Time Series, Area-Averaged” and the coordinates (Bounding Box and Shape) were delineated at 9.57°W, 39.35°N, and 9.45°W, 39.46°N, the area around the colony, defining an area approximately 12 x 11 km in size around the colony. This area represents the core foraging grounds used by YLGs from Berlenga Island at sea [22,40].’ 

Additionally, we have included the distance of the colony from the coast (L167-169) ‘Berlenga is a small rocky neritic island of ca. 78.8 ha, located within the continental shelf about 11 km off western Portugal coast, surrounded by shallow waters.’

L272-280. Why not on a monthly basis? Where is the landfill in relation to the breeding colony? What do you mean 3 potential food sources?

Reply: The phrase ‘three potential food sources’ has been removed from the sentence to avoid confusion. The sources are now explicitly stated in this section as: the annual weight of residues delivered to the proximate active landfill, the annual weight of non-differentiated residues collected in the city of Peniche and for the broader West region. This information was not available in a monthly basis. Additionally, the landfill is located approximately 40 km southeast from the colony, and the distance and direction are now indicated.

L281.314. The stats section needs clarity and needs to be completely re-written. Currently, this is impossible to determine what has been done. Please refer back to your aims and ensure that the analyses are presented in the order of the aims. At the moment, we have to fish for information in every sections. I also don’t think that Pearson’s correlations are correct. You have yearly and monthly effects, so you need to consider time series analyses. What is SIA and which variables were obtained form this? Limit abbreviations as much as possible as this makes it difficult to remember them all.

Reply: The data analysis section has been rewritten to align with the re-analysis we conducted. We have transitioned to using Linear Models (LMs) to ensure the robustness of our data analyses. This revised section now offers greater clarity regarding the specific procedures that were carried out. Furthermore, with regard to the abbreviation "SIA," while it is explained in detail in Section 2.3, we understand the importance of minimizing the use of abbreviations to enhance readability. To provide additional clarity and detail on these metrics, we remind the reader here that "SIA" stands for stable isotope analysis and specify SIA variables accordingly ‘SIA variables (δ15N and δ13C values, TA, SEAc and SEA.B)’.

Results

You need to report the statistical results even for non-significant relationships or refer to tables presenting the statistical results. Following on my comments regarding the statistical analyses, it also appears like you have run separate correlations while you need to combine all factors into the same analysis. I have not commented further on the results as the statistics need to be completely re-done.

Reply: We have re-conducted the statistical analysis as suggested. In the revised analysis, we utilized Linear Models (LMs) to comprehensively analyse our dataset, incorporating all relevant factors into a unified analysis. The results and methods sections have been subsequently revised to enhance clarity and detail. Additionally, we now include statistical results for non-significant relationships as well. We hope these improvements better address your concerns.

The sample size for each steps need to be clearly presented. At the moment, it is unclear how many individuals you sampled from and thus the representativeness of the results. From the methods, it appears as if you only sampled individuals in the control zone. Is this correct? This would represent only a small number of individuals (although, I’m only guessing here – see my comment above re: L159). How did you ensure that you did not resample the same individuals across years? Same with the number of eggs measured? How many eggs were measured and from how many pairs?

Reply: The sample sizes for each step have been clearly presented, as detailed in the performed analyses and stated in the methods section. We have explicitly mentioned in the methods section that individuals were not resampled (birds were always ringed) and the sampling of birds occurred across the entire island with the exception of control areas as stated in L218-219 ‘The nests were chosen randomly across the entire island (excluding control areas), and individuals were not resampled across years.’ Furthermore, the information regarding the measurement of the eggs is provided in L206-208 ‘Additionally, to ascertain mean egg volume of the clutch (egg volume), only nests with three eggs (the dominant class size) were randomly selected (sample size varied from 20 to 30 nests) in the control areas since 2012 (except in 2013 and 2020).’

Discussion

The discussion will need to be re-written once the results have been re-analysed. In general, however, the discussion is too narrowly focused on the study species and needs to be broadened.

Reply: The discussion has been enhanced to align with the re-analyses conducted, broadening its scope and making it more precise and concise. It's worth noting that in several aspects, the new analyses did not significantly deviate from previous ones and thus did not substantially modify our main conclusions. However, some sections underwent substantial changes, especially regarding the relationships between breeding performance and fisheries landings, and these changes were thoroughly addressed in the discussion.

MINOR COMMENTS

Abstract

L15. “There was exponential growth of the population until 1994” should be changed to “The population grew exponentially”.

Reply: Changed as suggested.

L26. “…at a faster rate”. Faster than what?

Reply: This part was removed from the abstract.

Introduction

L33. What impacts?

Reply: This sentence has been improved to clarify the impacts, and it now reads as follows (L44-46) ‘…prompting concern due to the impacts caused on ecosystems and urban areas, such as changes in vegetation cover and plant composition, competition with other species or interaction with humans [9–13].’

L44-47. Ok but what were the results of these investigations?

Reply: We have summarized the principal findings of these investigations in L52-57 ‘These studies have revealed significant insights into the foraging behaviour, dietary preferences, and ecological roles of gulls within their respective habitats, such as adaptations to changes in prey availability. Additionally, these investigations have provided valuable insights into the potential role of gulls as bioindicators to assess environmental shifts and anthropogenic influences on coastal ecosystems.’

 L52-54, L74-75, L85-86. You need some references here.

Reply: References added.

 L67. How much of the population did this represent?

Reply: This sentence was clarified accordingly (L80-82) ‘…with the exception of two designated control areas, representing less than 1% of the population [36].’

 L79. In which way?

Rely: The sentence was clarified and now reads as (L93-95) ‘Yet, longer trips might negatively influence their breeding performance, considering the impact of increased foraging time on the time spent at the nest [41].’

 L126. Please specify which specific parameters, data and records were obtained. What do you mean by fisheries landings (L132)?

Reply: The parameters have been specified, and fisheries landings have been addressed in the methods section with more detail.

L136-138. Unnecessary. I would replace however with some predictions.

Reply: This part was removed as suggested, and predictions have been included as suggested (L155-162) ‘We hypothesize that YLG population dynamics on Berlenga Island are influenced by variations in environmental factors, despite implemented population control measures. We also predict that the reproductive performance and trophic niche of YLGs will be influenced by the quality of oceanographic conditions, with better conditions associated with higher hatching success and narrower trophic niches. This presumes that changes in environmental factors, such as oceanographic conditions, fisheries landings, and urban waste can affect the annual reproductive performance and trophic ecology of the gull population.’

Methods

L142. Data is always plural. Please correct here and through the manuscript. Which specific data? The bird data or all data?

Reply: Corrected. We mean bird data, corrected.

 L167-169. Why is this information important? Re-state.

Reply: This part has been clarified, and the importance of the information has been restated as follows (L211-214) ‘The breeding parameters (total eggs pair-1, hatching success, first egg and egg volume) were used as measures of the gulls’ breeding performance during different years in relation to their trophic ecology, as well as oceanographic conditions, fisheries landings and urban waste.’

 L173-176. This information should be moved to 2.3

Reply: This information regarding BMI was removed from the manuscript.

L177-181. This information should be moved to 2.4.

Reply: Moved as suggested.

 L183. But you said above that no bird were captured in 2019.

Reply: The information regarding body mass index (and thus 2019) was excluded from the study to enhance the precision and conciseness of the manuscript and to prevent any possible misinterpretations. This was because body mass index was not a primary focus in the main objectives of this study, and no solid conclusions were drawn.

 L185-187. Please break this sentence into smaller sentences. Statistics and variables used for the analyses should be clearly stated and moved to the stats section.

Reply: This sentence was shortened and the statistics were moved to the data analyses section as suggested.

 L204-220. If you are using standard methods for this analysis, you should add some references.

Reply: Two references were added in this section where appropriate.

 L152. What do you mean by fisheries landings? Number of fishes caught?

Reply: The meaning of "fisheries landings" has been clarified (L292-294) ‘Data on fisheries landings (traded at fish auction, in kg, by species, year and month) were obtained from the main fishing harbour in the designated area, Peniche harbour (Figure 1)’.

 L264-265. Specify which species were allocated to which group.

Reply: The pelagic and demersal species are now specified in the text (L297-303) ‘… the pelagic European pilchard (Sardina pilchardus), Atlantic mackerel and Atlantic chub mackerel (Scomber spp.), Atlantic horse mackerel, blue jack mackerel and Medi-terranean horse mackerel (Trachurus spp.) , garfish (Belone belone), and the demersal blue whiting (Micromesistius poutassou), bogue (Boops boops), European conger (Conger conger), zebra seabream, sheephead bream, white seabream and common two-banded seabream (Diplodus spp.), pouting (Trisopterus luscus) and European hake (Merluccius merluccius).’

 L266-227. Unclear.

Reply: Background information has been provided throughout the text to clarify the concept of fisheries landings, particularly in the paragraph in question.

Discussion

Why do you switch to present tense in the discussion?

Reply: Corrected to past tense.

What is the return rate in the population? Do you have any information on survival rate and recruitment? This should be discussed or at least acknowledged.

Reply: Available information on recruitment, supported with appropriate references, has been added (L541-544) ‘The observed decrease in the density of nests in control areas suggested that there was no influx of breeding individuals. Even if there was an influx of migrants from other colonies [35], it would not have been sufficient to compensate for the number of breeding individuals that had perished [36].’

Reviewer 2 Report

Comments and Suggestions for Authors

I have completed my review for ‘Gulls as Indicators of Environmental Changes in the North Atlantic: A Long-Term Study on Berlenga Island, Western Portugal’, which is currently under consideration for publication in Diversity. Here, the authors explored the relationship between environmental factors, trophic ecology, number of breeding birds, and reproductive success of a large colony of yellow-legged gulls on the island.

There were exciting results. Although oceanographic conditions and fisheries landing influenced the trophic ecology of gulls, their overall population size and breeding performance were not affected. The gull population is subject to population control measures more relevant to the population size.  The results were interesting. While the trophic ecology of gulls was affected by oceanographic conditions and fisheries landing, their overall population size and breeding performance remained unaffected. The population of gulls is managed with population control measures that are more relevant to the size of the population. However, there was evidence of the effect of poor environmental conditions on the trophic ecology and population size.

I enjoyed reading the manuscript and believe it could be essential to assessing the impact of environmental conditions and human activities on seabirds, particularly coastal gull species with unique life histories. However, the manuscript has some issues that need further clarification.

More information about the sampling procedures is needed. In particular, the response variables of the breeding performance metrics (total egg pair, first egg, egg volume, and hatching success) were collected from the colony and control areas. Moreover, some samples were collected from nests with three eggs. For example, the egg volume (lines 167-168), body mass, and blood samples (lines 173-181). This specific sampling may influence the data because these nests/individuals may represent individuals with particular attributes/capacities instead of sampling individuals with different clutch sizes. It is possible that more/less experienced individuals respond to environmental conditions differently and then use various trophic resources.

The environmental variables may be correlated, for example, surface Chl-a and SST. This collinearity may result in confusing statistical analyses. Also, there is potential for spatial and temporal correlations, particularly in the relationships between trophic ecology and oceanographic parameters. The authors should consider these issues in their analyses and discussion.

The isotopic niches indicate larger niches in 2011, 2013, and 2016. However, there was no relationship between fisheries landings and urban waste. Also, it is confusing that there is a negative correlation between total eggs per pair and the mean δ15N value of RBC. According to the sampling procedures, blood samples were collected from individuals with three eggs, see above comments.

The discussion needs to be precise and concise. An exponential increase between 1974 and 1994 in the breeding population size was unrelated to the wNAO. Alternative available prey (e.g., discards from fisheries) played a critical role in this population increase. However, after the control population measures were implemented, the population dynamics were unrelated to the environmental conditions. What the gulls forage on is still a function of the environmental conditions.

A minor comment: please double-check the text because there are some references (e.g., Post et al. 2007, line 211).

Author Response

Thank you for reviewing our manuscript and for suggesting improvements. We are grateful for the careful reviews and a revision of the ms. was performed as recommended. We have revised the respective parts of the ms. following the helpful comments, particularly in the introduction and methods sections. The text has been improved, and we conducted a re-analysis using linear models (LMs) to ensure robust results. We are now submitting a revision of the manuscript addressing, point-by-point, the issues raised in the comments.

Comments and Suggestions for Authors

I have completed my review for ‘Gulls as Indicators of Environmental Changes in the North Atlantic: A Long-Term Study on Berlenga Island, Western Portugal’, which is currently under consideration for publication in Diversity. Here, the authors explored the relationship between environmental factors, trophic ecology, number of breeding birds, and reproductive success of a large colony of yellow-legged gulls on the island.

There were exciting results. Although oceanographic conditions and fisheries landing influenced the trophic ecology of gulls, their overall population size and breeding performance were not affected. The gull population is subject to population control measures more relevant to the population size.  The results were interesting. While the trophic ecology of gulls was affected by oceanographic conditions and fisheries landing, their overall population size and breeding performance remained unaffected. The population of gulls is managed with population control measures that are more relevant to the size of the population. However, there was evidence of the effect of poor environmental conditions on the trophic ecology and population size.

I enjoyed reading the manuscript and believe it could be essential to assessing the impact of environmental conditions and human activities on seabirds, particularly coastal gull species with unique life histories. However, the manuscript has some issues that need further clarification.

More information about the sampling procedures is needed. In particular, the response variables of the breeding performance metrics (total egg pair, first egg, egg volume, and hatching success) were collected from the colony and control areas. Moreover, some samples were collected from nests with three eggs. For example, the egg volume (lines 167-168), body mass, and blood samples (lines 173-181). This specific sampling may influence the data because these nests/individuals may represent individuals with particular attributes/capacities instead of sampling individuals with different clutch sizes. It is possible that more/less experienced individuals respond to environmental conditions differently and then use various trophic resources.

Reply: Thank you for your constructive feedback on the manuscript. We have made several adjustments based on reviewer’s suggestions. In the methods section, we have provided more detailed information regarding the sampling procedures, specifically concerning the response variables for breeding performance metrics. Egg volume measurements were limited to nests with three eggs. This decision was made because we observed a consistent decrease in egg size in the final laid eggs. Additionally, these measurements were exclusively taken from control areas, ensuring that they were unequivocally from the first clutch. This approach was chosen to facilitate year-to-year comparisons without introducing bias. As for the selection of nests with three-egg clutches for deploying traps and to sample individuals, this decision was made for practical reasons, as it made capturing and recapturing the gulls easier (in some gulls we have proceed to deploy GPS loggers, not used in the present study). While we acknowledge that this specific sampling strategy may introduce bias by focusing on individuals with particular attributes or capacities, it allowed us to standardize our sample in terms of the timing of the individual effort. It is worth noting that the majority of breeding pairs typically lay three-egg clutches, and their asynchronous nesting behaviour makes it challenging to predict whether clutches will consist of only one or two eggs.

The environmental variables may be correlated, for example, surface Chl-a and SST. This collinearity may result in confusing statistical analyses. Also, there is potential for spatial and temporal correlations, particularly in the relationships between trophic ecology and oceanographic parameters. The authors should consider these issues in their analyses and discussion.

Reply: We have re-conducted the statistical analysis as suggested. In the revised analysis, we utilized Linear Models (LMs) to comprehensively analyse our dataset, incorporating all relevant factors into a unified analysis. The results and methods sections have been subsequently revised to enhance clarity and detail. We hope these improvements better address your concerns.

The isotopic niches indicate larger niches in 2011, 2013, and 2016. However, there was no relationship between fisheries landings and urban waste. Also, it is confusing that there is a negative correlation between total eggs per pair and the mean δ15N value of RBC. According to the sampling procedures, blood samples were collected from individuals with three eggs, see above comments.

Reply: In fact, there were no significant relationships found between SEAc values (i.e., niche width) and fisheries landings or urban waste, which has been confirmed through a re-conducted statistical analysis that yielded consistent results with our previous findings. This suggests that the observed larger niches may be a consequence of unfavourable oceanographic conditions, leading to reduced specialization in marine resources such as the Henslow's swimming crab Polybius henslowii, as discussed, rather than being directly influenced by the exploitation of anthropogenic resources. However, it's worth noting that we did identify significant relationships between δ15N values and fisheries landings, particularly in the case of demersal carnivorous species (e.g., Diplodus spp., Merluccius merluccius), which are more likely to exhibit higher δ15N values.

The total number of eggs per breeding pair was estimated based on information collected during the annual census and population control campaigns, calculated as follows: total eggs pair-1 = total eggs / (number of breeding individuals / 2). This estimation is closely related to the overall population dynamics and is not directly correlated with the gulls that were sampled for the trophic ecology study. We have now provided additional details about this calculation in the methods section for clarity.

The discussion needs to be precise and concise. An exponential increase between 1974 and 1994 in the breeding population size was unrelated to the wNAO. Alternative available prey (e.g., discards from fisheries) played a critical role in this population increase. However, after the control population measures were implemented, the population dynamics were unrelated to the environmental conditions. What the gulls forage on is still a function of the environmental conditions.

Reply: We appreciate the reviewer's observation that the gulls' foraging behaviour is still influenced by environmental conditions even after the implementation of control population measures. Undoubtedly, discards from fisheries continue to play a significant role in the trophic ecology of this population, as demonstrated in several studies (e.g., Ceia et al., 2014; Alonso et al., 2015; Mendes et al., 2018; Calado et al., 2020; Oliveira et al., 2023) and had a crucial impact on the exponential population increase between 1974 and 1994. We have clarified and enhanced this portion of the discussion, including additional information related to wNAO and fisheries, and in line with the re-analyses performed.

A minor comment: please double-check the text because there are some references (e.g., Post et al. 2007, line 211).

Reply: Corrected.